# Lower Urinary Tract Symptoms (LUTS) as a New Clinical Presentation of Histamine Intolerance: A Prevalence Study of Genetic Diamine Oxidase Deficiency

**DOI:** 10.3390/jcm12216870

**Published:** 2023-10-31

**Authors:** Jose Ponce Díaz-Reixa, Marcos Aller Rodríguez, Sara Martínez Breijo, Jorge Suanzes Hernández, Eva Ruiz Casares, Teresa Perucho Alcalde, Manuel Bohorquez Cruz, Teresa Mosquera Seoane, Jose M. Sánchez Merino, Jacobo Freire Calvo, Paula Fernández Suárez, Venancio Chantada Abal

**Affiliations:** 1Urology Department, Complexo Hospitalario Universitario A Coruña (CHUAC), 15006 A Coruña, Spain; marcos.aller.rodriguez@sergas.es (M.A.R.); sara.martinez.breijo@sergas.es (S.M.B.); manuel.alejandro.bohoruqez.cruz@sergas.es (M.B.C.); teresa-mosquera.seoane@sergas.es (T.M.S.); jose.maria.sanchez.merino@sergas.es (J.M.S.M.); jacobo.freire.calvo@sergas.es (J.F.C.); venancio.chantada.abal@sergas.es (V.C.A.); 2Statistics Department, Complexo Hospitalario Universitario A Coruña (CHUAC), 15006 A Coruña, Spain; jorge.suanzes.hernandez@sergas.es; 3VIVO Laboratorio, Grupo Vivo, 28100 Alcobendas, Spain; rcasares@ceu.es (E.R.C.); teresaperucho@gmail.com (T.P.A.); 4Department of Genetics, Faculty of Medicine, CEU-San Pablo University, 28668 Madrid, Spain; 5Faculty of Biological Sciences, Complutense University of Madrid, 28040 Madrid, Spain; 6Radiodiagnostic Department, Complexo Hospitalario Universitario A Coruña (CHUAC), 15006 A Coruña, Spain; paula.fernandez.suarez@sergas.es

**Keywords:** lower urinary tract symptoms, histamine intolerance, single nucleotide polymorphisms, diamine oxidase, *DAO* gene

## Abstract

Lower urinary tract symptoms (LUTS) are highly prevalent, and their treatment is mainly focused on the control of symptoms. Histamine intolerance (HIT) has been related to a variety of systemic symptoms. DAO deficiency has been identified as a significant factor contributing to histamine intolerance (HIT). Preclinical evidence indicates the involvement of histamine in the lower urinary tract. This study aimed to assess the prevalence of diamine oxidase deficiency (*DAO*) in a prospective cohort of 100 patients with at least moderate LUTS. A genetic study of four single nucleotide polymorphisms (SNPs) (c.-691G>T, c.47C>T, c.995C>T, and c.1990C>G) was performed. HIT was found in 85.9% of patients. The prevalence of at least one minor allele in the SNPs analyzed was 88%, without gender differences. Storage symptoms were more intense in the presence of HIT as well as asthenia and neurological and musculoskeletal symptoms. The presence of minor alleles of the *AOC1* gene was associated with a higher intensity of symptoms. Minor alleles from c.-691G>T and c.47C>T SNPs were also associated with a greater severity of obstructive symptoms. Thirty-one percent of patients presented the four SNPS with at least one associated minor allele. The relationship between HIT and LUTS in a mixed population of men and women found in this study supports further investigations to define the pathophysiology of histamine in LUTS.

## 1. Introduction

Lower urinary tract symptoms (LUTS) (urgency, frequency, voiding difficulty, and/or incontinence) have a high prevalence and a negative impact on the quality of life [1]. It has usually been considered that these symptoms are associated with advanced age and most studies and clinical practice guidelines are addressed to this segment of the population, although many young patients also suffer from LUTS. In the EPIC study, the prevalence of LUTS increased with age, from 51% in the 18–39 year group, 62% between 40 and 59 years, and up to 80% in patients older than 60 years [1]. In young people, filling symptoms (37.5%) were more frequent than voiding symptoms (19.9%) [1]. Pharmacological treatments are effective in a percentage of patients, but the control of symptoms is limited by both adverse effects and adherence to treatment [2,3,4]. The healthcare costs of LUTS is high [5,6] with significant growing trends in the consumption of human and economic resources due to the increase in the prevalence of LUTS and aging of the population [7]. Up to the present time, LUTS have not been evaluated in the clinical context of intolerance to histamine.

Histamine intolerance (HIT) is a relatively recently described medical condition that has gained increasing attention for explaining a large variety of systemic symptoms. HIT is influenced by an imbalance between the ingestion of histamine-rich foods and the intestinal capacity of histamine degradation by the diamine oxidase (*DAO*) enzyme [8]. To date, LUTS have not been reported among the clinical manifestations of HIT. *DAO* is an enzyme encoded by the amine oxidase copper-containing 1 (*AOC1*) gene and responsible for extracellular degradation of histamine at the intestinal level [9,10,11]. It has been seen that the presence of minor alleles in some single nucleotide polymorphisms (SNPs) is associated with lower enzymatic activity of *DAO* and a greater expression of HIT symptoms [12,13,14]. The three most relevant SNPs in Caucasian individuals leading to a reduction in *DAO* enzyme activity are c.47C>T (rs10156191), c.995C>T (rs1049742), and c.1990C>G (rs1049793) [12,13]. The following frequencies (95% confidence interval [CI]) for these three *AOC1* variants among Spanish Caucasian individuals have been reported: rs10156191, 25.4% (20.16–30.58); rs1049742, 6.3% (3.42–9.26); and rs1049793, 30.6% (25.1–36.1) [12]. In addition, another SNP in the promoter region of the *AOC1* gene has been identified, with a frequency of 41.7%, which has been associated with a decrease in *DAO* transcriptional activity: c.-691G>T (rs2052129) [14].

Histamine and its metabolites are excreted through the urinary tract and increased urinary excretion of histamine has been shown during migraine attacks [15] or atopic dermatitis in children [16]. In addition, histamine urinary excretion can be used as a marker of plasma histamine levels and correlates with systemic effects of this compound [17]. In 1969, Todd and Mack [18] originally reported a contractile role of histamine in human bladder detrusor smooth muscle, which was not inhibited by anticholinergic agents and considered histamine as a non-specific smooth muscle stimulant. Subsequently, the University of Pennsylvania’s research team showed that the smooth muscle in the urinary bladder displayed an escalating reaction to histamine stimulation. This response involved contraction in the body of the bladder at lower doses, followed by the trigone, and ultimately, the bladder neck at progressively higher doses. This would account for predominantly filling symptoms with low histamine doses and voiding symptoms with higher doses. It was already suspected that contraction was mediated by H1 receptors, independently of acetylcholine. Khanna et al. [19] proposed that histamine should be considered as a new neurotransmitter of the urothelium, so that any inflammatory and/or allergic process could cause symptoms of the lower urinary tract [19].

In the 1980s, Barker and Ebersole [20] showed that histamine potentiates the release of other contractile substances in human muscle tissues, mainly acetylcholine and that this effect was mediated by H2 receptors. At the end of the 1980s, it was already found that histamine showed a high sensitivity contractile response (at low doses) mediated by H2 receptors that could be blocked by atropine or ranitidine, and an effect on cholinergic receptors, as well as an independent low sensitivity contractile response (at high doses) mediated by H1 receptors, which could be blocked by H1 antagonists [21]. Moreover, it was found that histamine has a stimulatory function through the purinergic system, mediated by the release of ATP at the post-synaptic level [22]. All these findings make the pharmacological management of LUTS symptoms very complex, since anticholinergic and antihistamine drugs would be needed for the blockage of both receptors.

In cultured smooth muscle cells from human detrusor biopsies in interstitial cystitis, Neuhaus et al. [23] showed that histamine and its agonists induced an intracellular calcium release response and that this action was inhibited by H1 and H3 blockers. In a comparison of the histamine receptor expression in cultured smooth muscle cells from the human detrusor and the internal sphincter [24], histamine stimulated significantly more cells than carbamylcholine (an acetylcholine receptor agonist) in detrusor and sphincter cells; however, histamine and specific agonists stimulated more sphincter cells than did carbachol and the calcium increase was greater in sphincter cells than in detrusor cells. The authors suggested that some of the variability in the outcome of antihistaminic interstitial cystitis therapies might be caused by the ineffectiveness of the chosen antihistaminic or unintentional weakening of sphincteric function [24]. A subsequent study illustrated the complexity of managing urinary tract clinical manifestations, because of the overexpression of different receptors, such as H1 and H2 as well as muscarinic and purinergic receptors [25]. On the basis of individual upregulated receptors profile, tailored therapy by specific receptor inhibitors may be promoted [25].

In recent years, there has been an increasing interest in the role of the urothelium and lamina propria in the modulation and regulation of the overall contractile capacity of the bladder. It has been shown that the urothelium and lamina propria are capable of releasing substances in response to various chemical and/or mechanical stimuli, such as acetylcholine, which can induce detrusor contraction [26]. On the other hand, mast cells play a central role in inflammatory and immediate allergic reactions and are able to release potent inflammatory mediators, such as histamine, chemotactic factors, cytokines, and proteases that act on smooth muscle, connective tissue, mucous glands, and other immune cells [27]. In bladder biopsy specimens from patients with interstitial cystitis, mast cells were increased in the urothelium, lamina propria, and smooth muscle [28]. In a histological study of 179 biopsies from patients with overactive bladder, signs of chronic inflammation were present in 69.8% of the cases (in the lamina propia in 98.4% of biopsies and in the urothelium in 17.6%) [29]. In a study of the effect of histamine on porcine urinary bladder samples, stimulation of H1 receptors resulted in contractions in both urothelium with lamina propria and detrusor tissue samples, whereas activation of the H2 receptor inhibited the H1-mediated contractions in the urothelium with lamina propria but not the detrusor, with H3 and H4 receptors having no functional role in bladder contractility [30]. In a mouse ex vivo bladder afferent preparation, it has been shown that intravesical histamine caused enhanced mechanosensitivity to bladder distention, and in conditions such as interstitial cystitis/bladder pain syndrome where mast cells and urinary histamine concentrations are increased, the contribution of histamine to sensitization of mechanosensitive bladder afferent pathways results in symptoms of urgency, frequency, and suprapubic pain at physiological distension pressures [31].

Taking into account these previous scientific advancements, it is crucial to investigate the role of *DAO* deficiency in LUTS. The objective of this study was to assess the prevalence of genetic *DAO* deficiency, considered as the presence of at least one minor allele in the SNPs of the *AOC1* gene (rsrs2052129, rs10156191, rs1049742, and/or rs1049793) in patients with LUTS.

## 2. Materials and Methods

### 2.1. Study Design

A prospective cohort study was carried out at Complexo Hospitalario Universitario A Coruña (CHUAC) in A Coruña, Galicia, Spain. The aim of the study was to assess the association between genetic *DAO* deficiency and LUTS in a mixed population of men and women with urinary symptoms.

### 2.2. Study Population

Between October 2022 and January 2023, a total of 100 adults of both sexes presenting with at least moderate LUTS were recruited. The inclusion criteria comprised individuals over 18 years old, with a minimum score of 4 points on the Bladder Control Self-Assessment Questionnaire (B-SAQ) or a minimum score of 8 points on the International Prostate Symptom Score (IPSS) questionnaire, in addition to providing signed informed consent.. Exclusion criteria were as follows: pharmacological treatment for LUTS (anticholinergics, antimuscarinics, and/or betamimetics) in the previous 2 weeks; history of bladder and/or prostate surgery; bladder tumor and/or intravesical chemotherapy in the previous 3 months; active urinary tract infection in the previous 2 weeks; history of colorectal surgery and/or abdominoperineal resection; spinal cord trauma, and current treatment with antihistamine drugs.

The study was approved by the Ethics Committee for Clinical Research (CEIC) of the Galician Health Service (code 2022/294 *DAO*-LUTS, approval date 13 June 2022). Written informed consent was obtained from all participants. The study was registered on 22 December 2022, with the ClinicalTrials.gov Identifier: NCT05676346.

### 2.3. Lower Urinary Tract Symptoms and Questionnaires

The intensity of LUTS was evaluated using the B-SAQ instrument for symptoms of overactive bladder and the IPSS questionnaire of prostatic symptoms. The B-SAQ questionnaire consists of four questions on symptoms of overactive bladder (urgency, frequency, nocturia, and urinary incontinence) included in two sections to measure the intensity and discomfort of symptoms. Symptoms are rated using a 4-point Likert scale (0 = not at all, 1 = a little, 2 = quite a lot, 3 = a lot) with a total range from 0 to 16. The final total score of the B-SAQ ranges between 0 and 32. A Spanish validated version of the B-SAQ was used [32].

The IPSS questionnaire measures the frequency with which symptoms occur throughout the day. It consists of seven questions and one question related to the patient’s perceived quality of life. Each question is subdivided into six possible answers, not at all, less than one time in five, less than half the time, about half the time, more than half the time, and almost always. The seven questions relating to symptoms experienced in the last month include a feeling of incomplete bladder emptying, frequency of urination, intermittency of urine stream, urgency of urination, weak stream, hesitation, and waking at night to urinate. The total score ranges from 0 to 35. A score 0 to 7 indicates mild symptoms, 8 to 19 moderate symptoms, and 20 to 35 severe symptoms. Severe symptoms were defined as an IPSS score ≥ 20 and mid/moderate < 20. Symptoms can also be classified into storage symptoms (The S-IPSS subindex is determined by summing the scores from questions #2, #4, and #7) and voiding symptoms (V-IPSS subindex, questions #1, #3, #5, #6). A Spanish validated version of the IPSS was used [33].

### 2.4. Histamine Intolerance

HIT was defined on clinical grounds based on (a) the appearance of typical clinical manifestations affecting different organs (e.g., rhinorrhea, rhinitis, nasal congestion, sneezing, headache, dizziness, pruritus, flush, urticaria, eczema, tachycardia, hypotonia, bloating, flatulence, diarrhea, abdominal pain, constipation, nausea, postprandial fullness, etc.), (b) improvement/remission after excluding foods that contain high histamine levels, and (c) the exclusion of other related disorders or the use of active ingredients with an inhibitory effect on the *DAO* enzyme [8].

### 2.5. AOC1 Variant Genotyping

A genetic study was performed on four variants of the *AOC1* gene in each sample: c.691G>T (rs2052129), p.Thr16Met (rs10156191), p.Ser332Phe (rs1049742), and p.His664Asp (rs1049793). The samples were collected from oral mucosa using a sterile cotton swab, by rubbing the inner side of both cheeks, following the standard hygiene protocol, maintaining a clean mouth for 60 min before sample collection. One sample was collected from each patient. The samples were identified with a code, anonymously, and refrigerated until they are sent to the laboratory. An automatic isolation procedure using the QIASymphony SP platform (QIAGEN, Hilden, Germany) with QIASymphony DSP DNA Mini Kit (QIAGEN) was used for DNA isolation. Genotyping was performed with a Multiplex SNPE (Single Nucleotide Primer Extension) followed by capillary electrophoresis in an ABI 3500 Genetic Analyzer (Thermofisher Scientific, Applied Biosystems, Waltham, MA, USA). To obtain the most accurate results, methodological recommendations of the American College of Medical Genetics and Genomics and the Association for Molecular Pathology [34] were followed.

### 2.6. Statistical Analysis

Categorical variables are expressed as frequencies and percentages, and continuous variables as mean and standard deviation (SD). The chi-square test or the Fisher’s exact test were used for the comparison of qualitative variables, and the Student’s *t* test or the Wilcoxon test for the analysis of quantitative variables according to conditions of application. Odds ratio (OR) and 95% confidence intervals (CI) were estimated with the Haldane–Anscombe correction when necessary. A *p* value < 0.05 was considered statistically significant. The IBM SPSS Statistics 25 (IBM Corporation, Armonk, NY, USA) package was used for the analysis of data.

## 3. Results

### 3.1. Characteristics of the Study Population

The general characteristics of patients with LUTS are shown in Table 1. This was a mixed sample, with 46% of men, and women slightly younger than men (*p* = 0.011). Symptoms of overactive bladder according to the B-SAQ questionnaire were similar in both sexes and of moderate intensity. In relation to the IPSS questionnaire, differences in the total score between men and women were not found, although voiding symptoms were more predominant in men (*p* = 0.003). Symptoms were of moderate intensity in both men and women (58%) as well as regarding the subscales of filling (69%) and voiding (89%) symptomatology. There was a trend of a higher severity of filling symptoms in women (38.9% vs. 21.7%) as opposed to a higher severity of voiding symptoms in men (17.4% vs. 5.6%), although differences were not statistically significant.

### 3.2. DAO Deficiency

The prevalence of genetic *DAO* deficiency defined as the presence of at least one minor allele associated *DAO* deficient activity in the SPNs analyzed was 88% (91.3% in men and 85.2% in women, *p* = 0.808). It was found that 31% of patients presented the four SNPs analyzed with a minor allele, with a similar distribution in men and women. The distribution of allelic variants maintained the Hardy–Weinberg Equilibrium (Figure 1). As shown in Table 2, there was a high prevalence of genotypes with the presence of the minor allele, and in all of them homozygosity with the normal allele was the least prevalent: 39% vs. 61% in rs2052129 (c.691g>T); 34% vs. 66% in rs10156191 (p.Thr16Met); 43% vs. 57% in rs1049742 (p.Ser332Phe); and 41% vs. 59% in rs1049793 (p.His664Asp). The distribution by gender was similar except for a higher prevalence of rs1049793 (p.His664Asp) with the minor allele in women than in men (*p* = 0.046) (Table 2).

### 3.3. Histamine Intolerance and LUTS

Clinical criteria of HIT were recorded in 85.9% of patients, with digestive symptoms as the most frequent in 73.7% of the cases followed by neurological symptoms in 57.6%, otorhinolaryngological in 52.5%, musculoskeletal in 52.5%, dermatological in 48.5%, fatigue in 48.5%, cardiovascular in 37.4%, and allergic in 21.2%. A total of 70.7% of patients were taking medication for symptoms of HIT.

When evaluating LUTS and HIT symptoms, there were no statistical differences between the different symptoms and the total IPSS score. In the B-SAQ questionnaire, higher intensity of storage or irritative symptoms was observed in the presence vs. absence of neurological symptoms (mean B-SAQ score 12.7 [6.04] and 9.95 [6.16], respectively, *p* = 0.03), musculoskeletal symptoms (12.8 [6.07] and 10.1 [6.11], *p* = 0.025), and asthenia (13.1 [6.15] and 10.1 [5.97], *p* = 0.017). The diagnosis of HIT was also associated with greater storage symptomatology (12.1 [6.13] vs. 7.67 [5.41], *p* = 0.01).

### 3.4. LUTS and Genetic DAO Deficiency

Table 3 shows the mean scores of LUTS questionnaires (B-SAQ and IPSS) in the different SNPs. The presence of symptoms according to the B-SAQ questionnaire was unrelated to SNPs, whereas the mean scores of the IPSS questionnaire were significantly higher when the minor alleles rs2052129 (c.691G>T) and rs10156191 (p.Thr16Met) were present (18.8 [7.6] vs. 14.9 [5.1], *p* = 0.021 and 18.1 [7.7] vs. 15.3 [5.0], *p* = 0.047, respectively). In the other two minor alleles rs1049742 (p.Ser332Phe) and rs1049793 (p.His664Asp), differences in IPSS scores were not found.

When the intensity of symptoms (moderate vs. severe) according to the IPSS questionnaire was assessed (Table 4), female gender showed a protective effect, with a 58% lower risk of severe symptoms compared to males (45.7% vs. 25.9%; OR = 0.42, *p* = 0.029). In relation to the presence of alleles associated with *DAO* deficiency (either heterozygous or homozygous), the SNP rs2052129 (c.691G>T) showed a higher risk of severe symptoms (OR 2.48 (1.01–6.10), *p* = 0.046), as well as the co-occurrence of rs2052129 (c.691G>T) and rs10156191 (p.Thr16Met) (OR = 2.98 (1.21–7.33), *p* = 0.015) and the presence of all four SNPs with at least one minor allele (OR = 2.81(1.16–6.77), *p* = 0.020).

The distribution of SNPs genotypes according to the severity of LUTS in the storage and voiding subscales of the IPSS is shown in Table 5. Regarding storage symptoms, women showed a trend toward more severe symptoms than men, but female gender was not an independent variable significantly associated with allelic positivity. In relation to the voiding subscale, men showed more severe symptoms than women, but male gender was not a significant predictor of allelic positivity. However, the presence of the SNP rs2052129 (c.691G>T) and the rs10156191 (p.Thr16Met) as well as the co-occurrence of rs2052129 (c.691G>T) and rs10156191 (p.Thr16Met) were significantly associated with the severity of voiding symptoms. Also, the presence of the four SNPs and homozygosity were predictors of voiding symptoms severity (Table 5).

Differences between men and women according to the presence of severe voiding symptoms are shown in Table 6. Men had a higher risk of severe voiding symptoms in the presence of the SNP rs2052129 (c.691G>T) (*p* = 0.019) or the co-occurrence of rs2052129 (c.691G>T) and rs10156191 (p.Thr16Met) (*p* = 0.015), whereas women had a higher risk in the presence of the 4 SNPs (*p* = 0.018) and homozygosity of the minor allele (*p* = 0.030).

## 4. Discussion

This study shows a relationship between HIT and LUTS, a novel finding that has not been previously reported. Also, patients with LUTS presented a very high prevalence (88%) of at least one altered variant (presence of a minor allele associated with *DAO* deficiency) in the SNP of the *AOC1* gene. Although HIT has been commonly associated as a syndrome that predominantly affects women [14,35,36], our sample included 46% of men, suggesting that genetic *DAO* deficiency can affect both sexes. The studied alleles were consistent with Hardy–Weinberg equilibrium, but considering the database of SNF (dbSNP) that included NCBI Allele Frequency Aggregator (ALFA) data as a reference (https://www.ncbi.nlm.nih.gov/snp/docs/gsr/alfa/, accessed on 15 July 2023), the minor alleles show a higher incidence.

The prevalence of at least one altered SNP in the general population is not well known. However, in Caucasian people, it has been reported to be around 70% [12]. HIT caused by a deficiency in the *DAO* enzyme has been extensively investigated in recent years due to its involvement in multiple pathologies, such as migraine [35,37], fibromyalgia [36], inflammatory bowel disease [38], and hypersensitivity to non-steroidal anti-inflammatory drugs (NSAID) [39]. This study provides novel information that links LUTS with histamine metabolism.

Few studies have evaluated the prevalence of SNPs in the *AOC1* gene with *DAO* deficiency, mostly because of difficulties in the selection of a control population without HIT-related symptoms and normal *DAO* activity. Ayuso et al. [12] categorized three SNPs locations in the *AOC1* gene among 134 healthy Caucasian individuals and studied serum *DAO* activity in 37 of them. Variant ABP1 alleles leading to the amino-acid substitutions Thr16Met, Ser332Phe, and His645Asp were identified with frequencies of 25.4, 6.3, and 30.6%, respectively, and over 70% of the population carried at least one amino-acid substitution. These percentages in healthy individuals are lower than the allelic prevalence of 66% for Thr16Met, 57% for Ser332Phe, and 59% for His664Asp found in our study.

Maintz et al. [14] analyzed SNPs in 285 German patients with clinical symptoms of HIT and 199 controls. The minor allele at rs2052129, rs2268999, rs10156191, and rs1049742 increased the risk for a reduced *DAO* activity, and the reporter gene assays at rs2052129 revealed a significantly lower promoter activity of the minor allele. When *DAO* activity was severely diminished (*DAO* < 10 U/mL), symptoms of HIT were more significant compared to having *DAO* ≥ 10 U/mL. When comparing the prevalence of SNPs between patients with reduced *DAO* activity and those with normal *DAO* activity, it was observed that in patients with reduced *DAO* activity, the prevalence of the minor allele in c.691G>T (rs2052129) was 57.2%, similar to that found our series (61%). However, in patients with normal *DAO* activity, the prevalence of the same minor allele was only 31.4%, similar to the European population as shown in the ALFA dataset. A similar situation occurs in the SNP p.Thr16Met (rs10156191), where the prevalence of the minor allele is similar: 66% in the Mainz study [14] and 61% in our study, compared to 37.7% in patients with normal *DAO* activity (Figure 2). These data are consistent with the hypothesis that *DAO* activity allows the identification of populations with and without symptoms. In our study, serum *DAO* activity was not measured. However, the study population exhibits moderate symptoms of the lower urinary tract, and the prevalence of the minor allele of the *AOC1* gene is similar to that of patients with significantly reduced *DAO* activity (<10 U/mL) but a higher prevalence of HIT symptoms. Other studies have reported a lower prevalence of the minor allele in both SNPs [35,40]. It is important to note that the prevalence of the minor allele for these two SNPs (c.691G>T, rs2052129, and p.Thr16Met, rs10156191) in studies related to HIT symptoms and *DAO* genetic deficiency ranges from 28% to 57% in the first case and 31% to 46% in the second case [35,36,39,40]. These percentages are lower than the data observed in our series.

According to these data, the population of patients with reduced *DAO* activity would have a higher risk of presenting LUTS. Our study shows a very high prevalence, in both sexes, which has not been seen in previous prevalence studies of *DAO* deficiency. Symptoms of HIT are associated with increased severity of storage symptoms, especially in the presence of neurological and musculoskeletal symptoms and/or asthenia. Genetic *DAO* deficiency has been associated with a higher risk of severe obstructive symptoms. This finding has significant clinical implications, as obstructive symptoms encompass various conditions that require not only pharmacological but also surgical urological management. To date, there have been no studies published on clinical findings in patients with HIT, and previous publications have only shown preclinical evidence.

While acknowledging the limitations including a small sample size, the absence of control subjects without LUTS, and the non-measurement of serum DAO levels or urinary histamine excretion, it’s important to highlight a notable strength of the study. The research significantly contributes by shedding light on a new pathway in comprehending the etiology of LUTS, specifically by investigating a novel neurotransmitter within the urinary tract. 

## 5. Conclusions

This study shows a high prevalence of *DAO* genetic deficiency in patients with LUTS, both in men and women. The presence of HIT symptoms was associated with greater severity of storage symptoms. Also, the existence of minor alleles of the AOC1 gene demonstrated a significant correlation with the intensity of obstructive symptoms. These findings open up further research possibilities to better categorize the relationship between histamine and LUTS.

## Figures and Tables

**Figure 1 jcm-12-06870-f001:**
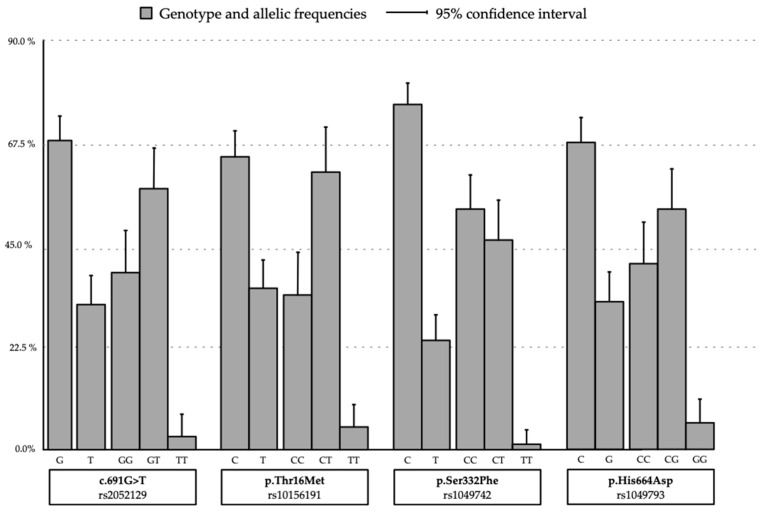
AOC1 SNPSs genotype and prevalence of allele variants.

**Figure 2 jcm-12-06870-f002:**
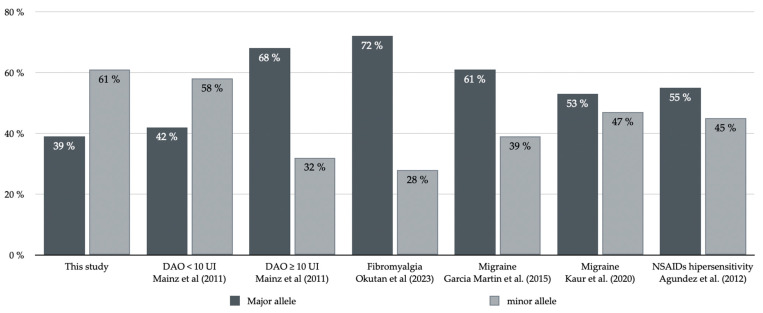
Prevalence of the SNP c691G>T (rs2052129) in published studies with control population and population with histamine intolerance [14,35,36,39,40].

**Table 1 jcm-12-06870-t001:** Clinical characteristics of the study population.

Variables	Total, *n* = 100	Men, *n* = 46	Women, *n* = 54	*p* Value
Age, years, mean (SD)	56.9 (14.6)	60.4 (13.4)	52.6 (14.7)	0.011
Age segments, *n* (%)				
<40 years	13 (13.0)	3 (6.5%)	10 (18.5%)	0.141
40–50 years	24 (24.0%)	9 (19.6%)	15 (27.8%)	
51–60 years	19 (19.0%)	9 (19.6%)	10 (18.5%)	
61–70 years	21 (21.0%)	10 (21.7%)	11 (20.4%)	
>70 years	23 (23.0%)	15 (32.5%)	8 (14.8%)	
B-SAQ score, mean (SD)	11.5 (6.2)	10.1 (6.3)	12.5 (5.9)	0.052
IPSS score, mean (SD)	17.2 (6.9)	18.6 (7.1)	16.0 (6.5)	0.074
S-IPSS score	8.0 (3.2)	7.6 (3.3)	8.3 (3.1)	0.174
V-IPSS score	9.2 (5.6)	11.0 (5.5)	7.6 (5.2)	0.003
IPSS severity, *n* (%)				
Mild	7 (7.0)	2 (4.3)	5 (9.3)	0.120
Moderate	58 (58.0)	23 (50.0)	35 (64.8)	
Severe	35 (35.0)	21 (45.7)	14 (25.9)	
S-IPSS severity, *n* (%)				
Moderate	69 (69.0)	36 (78.3)	33 (61.1)	0.065
Severe	31 (31.0)	10 (21.7)	21 (38.9)	
V-IPSS severity, *n* (%)				
Moderate	89 (89.0)	38 (82.6)	51 (94.4)	0.059
Severe	11 (11.0)	8 (17.4)	3 (5.6)	

B-SAQ: Bladder Control Self-Assessment Questionnaire; IPSS: International Prostate Symptom Score; S-IPSS: storage symptoms IPSS; V-IPSS: voiding symptoms IPSS; SD: standard deviation.

**Table 2 jcm-12-06870-t002:** Prevalence of SNPs in the *AOC1* gene in patients with LUTS.

Genetic Profile		Total Patients (*n* = 100)	Men (*n* = 46)	Women (*n* = 54)	*p* Value
SNPs minor allele, *n* (%)					
0		12 (12.0)	4 (8.7)	8 (14.8)	0.808
1		20 (20.0)	8 (17.4)	12 (22.2)	
2		22 (22.0)	11 (23.9)	11 (20.4)	
3		15 (15.0)	7 (15.2)	8 (14.8)	
4		31 (31.0)	16 (34.8)	15 (27.8)	
SNPs genotype, *n* (%)					
rs2052129 (c.691G>T)	GG	39 (39.0)	17 (36.9)	22 (40.7)	0.816
	GT	58 (58.0)	28 (60.9)	30 (55.6)	
	TT	3 (3.0)	1 (2.2)	2 (3.7)	
rs10156191 (p.Thr16Met)	CC	34 (34.0)	14 (30.4)	20 (37.0)	0.742
	CT	61 (61.0)	30 (65.2)	31 (57.4)	
	TT	5 (5.0)	2 (4.3)	3 (5.6)	
rs1049742 (p.Ser332Phe)	CC	43 (43.0)	24 (52.2)	29 (53.7)	0.912
	CT	56 (56.0)	22 (47.8)	24 (44.4)	
	TT	1 (1.0)	0	1 (1.1)	
rs1049743 (p.His664Asp)	CC	41 (41.0)	14 (30.4)	27 (50.0)	0.046
	CG	53 (53.0)	27 (58.7)	26 (48.1)	
	GG	6 (6.0)	5 (10.9)	1 (1.9)	

**Table 3 jcm-12-06870-t003:** LUTS and genetic *DAO* deficiency.

SNPs Genotypes		Prevalence	B-SAQ Score Mean (SD)	*p* Value	IPSS Score Mean (SD)	*p* Value
rs2052129 (c.691G>T)	GG	39%	11.3 (5.8)	0.900	14.9 (5.1)	0.021
	GT	58%	11.6 (6.4)		18.8 (7.6)	
	TT	3%	10.7 (10.7)		17.3 (3.1)	
rs10156191 (p.Thr16Met)	CC	34%	11.6 (5.9)	0.755	15.3 (3.0)	0.047
	CT	61%	11.2 (6.3)		18.1 (7.7)	
	TT	5%	13.2 (7.7)		20.8 (4.7)	
rs1049742 (p.Ser332Phe)	CC	43%	12.2 (5.8)	0.319	15.9 (6.1)	0.083
	CT	56%	10.6 (6.6)		18.5 (7.4)	
	TT	1%	14 (.)		28 (.)	
rs1049743 (p.His664Asp)	CC	41%	12.9 (6.0)	0.066	16.4 (6.2)	0.070
	CG	53%	10.9 (6.0)		17.2 (7.2)	
	GG	6%	6.5 (7.3)		23.7 (6.4)	

B-SAQ: Bladder Control Self-Assessment Questionnaire; IPSS: International Prostate Symptom Score; SD: standard deviation.

**Table 4 jcm-12-06870-t004:** Severity of LUTS according to the IPSS questionnaire and SNPs genotypes.

Variables		IPSS Score		Allele Positivity Odds Ratio (95% Confidence Interval)
		Mild/Moderate <20, *n* (%)	Severe ≥20, *n* (%)	
Gender	Male	25 (54.3)	21 (47.5)	0.42 (0.18–0.97); *p* = 0.039
	Female	40 (74.1)	14 (25.9)	
rs2052129 (c.691G>T)	GG	30 (76.9)	9 (23.1)	2.48 (1.01–6.10); *p* = 0.046
	GT-TT	35 (57.4)	26 (42.6)	
rs10156191 (*p*.Thr16Met)	CC	26 (76.5)	8 (23.5)	2.25 (0.89–5.71); *p* = 0.084
	CT-TT	39 (59.1)	27 (40.9)	
rs1049742 (p.Ser332Phe)	CC	38 (71.7)	15 (28.3)	1.88 (0.82–4.31); *p* = 0.136
	CT-TT	27 (57.4)	20 (42.6)	
rs1049743 (p.His664Asp)	CC	30 (73.2)	11 (26.8)	1.87 (0.79–4.44); *p* = 0.153
	CG-GG	35 (59.3)	24 (40.7)	
rs2052129 rs10156191	No	33 (78.6)	9 (21.4)	2.98 (1.21–7.33); *p* = 0.015
	Yes	32 (55.2)	26 (44.8)	
SNPs	<4 SNPs	50 (72.5)	19 (27.5)	2.81 (1.16–6.77); *p* = 0.020
	4 SNPs	15 (48.4)	16 (51.6)	
Homozygosity	No	59 (67.8)	38 (32.2)	2.46 (0.76–8.00); *p* = 0.210
	Yes	6 (46.2)	7 (53.8)	

IPSS: International Prostate Symptom Score; Allele positivity compares the presence of the minor allele (either in heterozygosity or homozygosity) related to reduced *DAO* activity vs. the absence of the minor allele.

**Table 5 jcm-12-06870-t005:** Severity of storage and voiding scores of the IPSS questionnaire and SNP genotype.

Variables		S-IPSS Score		Allele Positivity Odds Ratio (95% Confidence Interval)	V-IPSS		Allele Positivity Odds Ratio (95% Confidence Interval)
		Mild/ Moderate <20, *n* (%)	Severe ≥20, *n* (%)		Mild/ Moderate <20, *n* (%)	Severe ≥20, *n* (%)	
Gender	Male	36 (78.3)	10 (21.7)	2.29 (0.94–5.57); *p* = 0.065	38 (82.6)	8 (17.4)	0.28 (0.07–1.12); *p* = 0.059
	Female	33 (61.1)	21 (38.9)		51 (94.4)	3 (5.6)	
rs2052129 (c.691G>T)	GG	29 (74.4)	10 (25.6)	1.52 (0.62–3.71); *p* = 0.354	39 (100.0)	0	17.0 (1.03–314.7); *p* = 0.006 *
	GT-TT	40 (65.6)	21 (34.4)		50 (82.0)	11 (18.0)	
rs10156191 (p.Thr16Met)	CC	26 (76.5)	8 (23.5)	1.74 (0.68–4.45); *p* = 0.246	34 (100.0)	0	14.3 (0.82–250.4); *p* = 0.014 *
	CT-TT	43 (65.2)	23 (34.8)		55 (83.3)	11 (16.7)	
rs1049742 (p.Ser332Phe)	CC	38 (71.7)	15 (28.3)	1.31 (0.56–3.06); *p* = 0.536	50 (94.3)	3 (5.7)	3.42 (0.85–13.75); *p* = 0.070
	CT-TT	31 (66.0)	16 (34.0)		39 (83.0)	8 (17.0)	
rs1049743 (p.His664Asp)	CC	26 (63.4)	15 (36.6)	0.64 (0.27–1.52); *p* = 0.314	38 (92.7)	3 (7.3)	1.99 (0.49–7.99); *p* = 0.518
	CG-GG	43 (72.9)	16 (27.1)		51 (86.4)	3 (16.3)	
rs2052129 rs10156191	No	31 (73.8)	11 (26.2)	1.48 (0.62–3.56); *p* = 0.376	42 (100.0)	0	20.58 (1.86–359.9); *p* = 0.002 *
	Yes	38 (65.5)	20 (34.5)		47 (81.0)	11 (19.0)	
SNPs	< 4 SNPs	50 (72.5)	19 (27.5)	1.66 (0.68–4.07); *p* = 0.264	65 (94.2)	4 (5.8)	4.74 (1.27–17.65); *p* = 0.032
	4 SNPs	19 (61.3)	12 (38.7)		15 (48.4)	16 (51.6)	
Homozygosity	No	61 (70.1)	26 (29.9)	1.47 (0.44–4.90); *p* = 0.534	80 (92.0)	7 (8.0)	5.08 (1.24–20.77); *p* = 0.035
	Yes	8 (61.5)	5 (38.5)		9 (69.2)	4 (30.8)	

S-IPSS: storage subscale of the International Prostate Symptom Score; V-IPSS: voiding subscale of the International Prostate Symptom Score; * Estimated odds ratio with the Haldane–Anscombe correction or Fisher’s exact test.

**Table 6 jcm-12-06870-t006:** Voiding LUTS severity and SNP genotypes in men and women.

Variables		V-IPSS Score in Men		Allele Positivity *p* Value	V-IPSS in Women		Allele Positivity *p* Value
		Mild/ Moderate <17, *n* (%)	Severe ≥17, *n* (%)		Mild/ Moderate <17, *n* (%)	Severe ≥17, *n* (%)	
rs2052129 (c.691G>T)	GG	17 (44.7)	0	0.019	22 (43.1)	0	0.262
	GT-TT	21 (55.3)	8 (100.0)		29 (56.9)	3 (100.0)	
rs10156191 (p.Thr16Met)	CC	14 (36.8)	0	0.085	20 (39.2)	0	0.287
	CT-TT	24 (63.2)	8 (100.0)		31 (60.8)	3 (100.0)	
rs2052129 rs10156191	No	18 (47.4)	0	0.015	24 (47.1)	0	0.245
	Yes	20 (52.6)	8 (100.0)		27 (52.9)	3 (100.0)	
SNPs	<4 SNPs	26 (68.4)	4 (50.0)	0.421	39 (76.5)	0	0.018
	4 SNPs	12 (31.6)	4 (50.0)		12 (23.5)	3 (100.0)	
Homozygosity	No	33 (86.8)	6 (75.0)	0.597	47 (92.2)	1 (33.3)	0.030
	Yes	5 (13.2)	2 (25.0)		4 (7.8)	2 (66.7)	

Allele positivity compares the presence of the minor allele (either in heterozygosity or homozygosity) related to reduced *DAO* activity vs. the absence of the minor allele; significance level: chi-square test or Fisher’s exact test.

## Data Availability

Data of the study are available from the corresponding author upon request.

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
