# Peer review of "Lower Urinary Tract Symptoms (LUTS) as a New Clinical Presentation of Histamine Intolerance: A Prevalence Study of Genetic Diamine Oxidase Deficiency"

_jcm, 2023, doi:10.3390/jcm12216870_

Round 1

Reviewer 1 Report

Comments and Suggestions for Authors

The authors should be congratulated for their work and for addressing an important topic. Only some points warrant mention:

MAJOR COMMENTS:

1.    in the “Results” section, it would be really interesting if the authors could provide results by age, or at least if a regression model could analyse the relationship between LUTS in younger patients (<40 y.o.) and HIT. That could provide important implications in diagnosis and further therapies in younger patients seeking medical help for LUTS.

MINOR COMMENTS:

1.    in the “Introduction” section, line 44 needs a reference.

2.    in the “Materials and Methods” section, did the authors include/exclude patients with neurologic diseases?

Author Response

MAJOR COMMENTS:

1.    in the “Results” section, it would be really interesting if the authors could provide results by age, or at least if a regression model could analyze the relationship between LUTS in younger patients (<40 y.o.) and HIT. That could provide important implications in diagnosis and further therapies in younger patients seeking medical help for LUTS.

The study included a limited number of patients under the age of 40, accounting for only 13% of the sample. In this younger age group, the incidence of genetic DAO (Diamine Oxidase) deficiency was 92%, with all females affected and 92% of males. No statistical differences were observed compared to patients over 40 years old. 

The incidence of HIT (Histamine Intolerance) was 100% in all patients. This condition was distributed among various body systems: neurological (92%), digestive (77%), ORL (ear, nose, and throat - 54%), dermatological (69%), allergies (46%, with differences between younger and older patients), osteomuscular (69%), and fatigue (69%).

The assessment of Lower Urinary Tract Symptoms (LUTS) in young patients revealed the following findings:

In the B-SAQ questionnaire, a higher incidence of symptom severity was observed in adult females compared to young females (significant difference). Similar trends were noted in adult males, although statistical significance was not reached due to the small number of patients.

No statistical differences were observed in the categorization of significant symptoms (score of 3 or higher) in both males and females. These symptoms included incomplete voiding, frequency, intermitence, urgency (with differences between genders), weakness, straining, and quality of life.

Adult males experienced a higher severity of nocturia (2 or more episodes of nocturia) compared to young males.

When categorizing symptoms as obstructive or irritative, young patients showed a higher incidence of severe obstructive symptoms, while adults had a higher incidence of severe irritative symptoms (with significant differences in female patients).

Please note that the statistical significance and the number of patients in some subgroups may be limited due to the small sample size.

MINOR COMMENTS:

1.    in the “Introduction” section, line 44 needs a reference.

A reference has been included (Coyne, Sexton et al. 2009)

2.    in the “Materials and Methods” section, did the authors include/exclude patients with neurologic diseases?

Patients with neurological diseases were not excluded, but patients with neurogenic bladder secondary to spinal traumas were.

Reviewer 2 Report

Comments and Suggestions for Authors

It is well-written manuscript and provides a comprehensive overview of the potential link between histamine intolerance and symptoms of lower urinary tract. The article would have been more convincing and provide the direct evidence of the relationship between histamine intolerance and LUTS, if the urinary histamine level was determined and correlated with the LUTS, and with control subjects.

Comment:

In figure 1: The information about the error bars in the legend is missing. In order to make the graph easily comprehensible, the bars for each group should be clustered together. 

In figure 2, Instead of Ponce at al., it is better to write "This study".

Author Response

The article would have been more convincing and provide the direct evidence of the relationship between histamine intolerance and LUTS, if the urinary histamine level was determined and correlated with the LUTS, and with control subjects.

The current publication is an exploratory study of the prevalence of genetic DAO deficiency, aimed at determining whether there is a potential relationship between the systemic accumulation of histamine and urinary tract symptoms.

Based on these results, a sub-analysis is currently underway to examine 24-hour urine histamine levels, plasma DAO activity, and their correlation with the severity of urinary tract symptoms. These data will be published as soon as their results are available.

Comment:

In figure 1: The information about the error bars in the legend is missing. In order to make the graph easily comprehensible, the bars for each group should be clustered together. 

Corrected

In figure 2, Instead of Ponce at al., it is better to write "This study”.

Corrected